# Salivary Proteomics Markers for Preclinical Sjögren’s Syndrome: A Pilot Study

**DOI:** 10.3390/biom12060738

**Published:** 2022-05-24

**Authors:** Nicoletta Di Giorgi, Antonella Cecchettini, Elena Michelucci, Giovanni Signore, Elisa Ceccherini, Francesco Ferro, Elena Elefante, Chiara Tani, Chiara Baldini, Silvia Rocchiccioli

**Affiliations:** 1Clinical Physiology Institute-CNR, 56124 Pisa, Italy; antonella.cecchettini@unipi.it (A.C.); emichelucci@ifc.cnr.it (E.M.); ceccherini@ifc.cnr.it (E.C.); silvia.rocchiccioli@ifc.cnr.it (S.R.); 2Rheumatology Unit, Department of Clinical and Experimental Medicine, University of Pisa, 56126 Pisa, Italy; francescoferrodoc@gmail.com (F.F.); elena.elefante87@gmail.com (E.E.); chiaratani78@gmail.com (C.T.); 3Biochemistry Unit, Department of Biology, University of Pisa, 56126 Pisa, Italy; giovanni.signore@unipi.it

**Keywords:** primary Sjögren’s syndrome, preclinical Sjögren’s syndrome, autoimmunity, salivary proteomics, mass spectrometry

## Abstract

Primary Sjögren’s syndrome (pSS) is a complex autoimmune disorder that particularly affects the salivary and lachrymal glands, generally causing a typical dryness of the eyes and of the mouth. The disease encompasses diverse clinical representations and is characterized by B-cell polyclonal activation and autoantibodies production, including anti-Ro/SSA. Recently, it has been suggested that autoantibody profiling may enable researchers to identify susceptible asymptomatic individuals in a pre-disease state. In this pilot study, we used mass spectrometry to analyze and compare the salivary proteomics of patients with established pSS and patients with pre-clinical SS, identifying a common protein signature in their salivary fluid. We found that several inflammatory, immunity-related, and typical acinar proteins (such as MUC5B, PIP, CST4, and lipocalin 1) were differently expressed in pSS and in pre-clinical SSA+ carriers, compared to healthy controls. This suggests that saliva may closely reflect exocrine gland inflammation from the early phases of the disease. This study confirms the value of salivary proteomics for the identification of reliable biomarkers for SS that could be identified, even in a preclinical phase of the disease.

## 1. Introduction

Primary Sjögren’s syndrome (pSS) is a female-dominated, complex, and heterogeneous systemic autoimmune disorder that particularly affects the salivary and lachrymal glands, generally causing a typical dryness of the eyes and of the mouth [1]. The disease encompasses a wide spectrum of signs and symptoms and is characterized by B-cell polyclonal activation and autoantibodies production, including anti-Ro/SSA [2]. Generally, xerostomia and xerophthalmia are the leading symptoms in the diagnosis of pSS. However, the recent literature has shown that anti-Ro/SSA autoantibodies may be detected in the plasma of patients up to 18 years before the onset of clinical dryness [3]. It has been suggested, therefore, that autoantibody profiling may enable researchers to identify susceptible asymptomatic individuals in a pre-disease state. Recently, salivary proteomics has appeared as a promising tool for the recognition of pSS [4]. Indeed, proteomic has offered a unique opportunity to identify, non-invasively, certain salivary biomarkers able not only to differentiate pSS from no-SS sicca controls (where dry eyes and dry mouth are collectively described as “sicca symptoms”) but also to reflect salivary flow impairment or the intensity of immune cell infiltration in the tissue [5,6].

The direct use of whole saliva has, however, several disadvantages for proteomics analysis, including contamination and variability. Whole saliva contains various highly abundant proteins, such as amylase and immunoglobulin (IgG) that can mask other low-abundance proteins, like extracellular vesicle (EV)-associated proteins [7]. EVs are phospholipid bilayer membrane-enclosed structures, secreted and released by various cells, and are widely presented in various body fluids. EVs play a significant role in intercellular signaling; the EV-mediated transfer of biomolecules is a critical component of a variety of physiological and pathological processes [8]. EV proteomics has previously been evaluated as a tool for the identification of novel biomarkers in pSS [9]. It is likely that the concomitant analysis of whole saliva and EVs proteins may provide a comprehensive picture of the disease mechanisms, offering the possibility of identifying novel specific biomarkers [10].

In this regard, liquid chromatography, coupled with tandem mass spectrometry (LC-MS/MS) has become the technology of choice for the high-throughput characterization of proteins [11] and also, in recent years, for the identification of putative pSS salivary biomarkers [9,12]. Several different proteomic strategies have been developed; the sequential window acquisition of all theoretical mass spectra (SWATH-MS) [13] is a specific variant of data-independent acquisition (DIA) methods and is emerging as a technology that combines deep proteome coverage capabilities with quantitative consistency and accuracy [14].

This study fits in the novel field of interceptive medicine. Interceptive medicine is rapidly becoming a hot topic in different fields of medicine and is basically aimed at identifying clinically relevant biomarkers that may allow clinicians to recognize and treat disease at earlier time points during its course before the patient develops disease symptoms. In rheumatoid arthritis, great efforts have recently been made to facilitate research into the preclinical and earliest clinically apparent phases of the disease, emphasizing the crucial value of circulating autoantibodies, specifically the rheumatoid factor (RF) and/or anticitrullinated protein antibody (ACPA), as risk factors preceding the clinical onset of the disease [15].

In our study, we focused on preclinical Sjögren’s syndrome, a condition with anti-Ro/SSA autoantibodies that can be detected in the plasma of patients up to 18 years before the onset of the disease. Asymptomatic individuals showing a positive response for anti-Ro/SSA antibodies are, therefore, considered at risk of SS and are defined as patients in a preclinical stage of the disease. Individuals with anti-Ro/SSA antibodies are, therefore, likely to represent patients in important therapeutic windows, within which clinical outcomes can be significantly modulated.

Therefore, in this pilot study, we used SWATH analysis to investigate the proteomic biomarkers in anti-Ro/SSA asymptomatic patients (carrier SSA+) and we compared their salivary proteome with patients with fully expressed SS, in order to investigate the common early biomarkers that are potentially able to be translated into clinical practice for an early diagnosis of Sjögren’s syndrome.

## 2. Materials and Methods

### 2.1. Study Design, Patient Population and Ethical Approval 

Patients with pSS (*n* = 11) who fulfilled the ACR/EULAR 2016 criteria and carrier SSA+ subjects with preclinical SS (*n* = 8) were recruited from the outpatient cohort at the Rheumatology Clinic of the University of Pisa. All pSS patients presented a positive test for anti-Ro/SSA antibodies, complained of dry eyes and dry mouth, and presented reduced unstimulated salivary flow (mean 0.16 ± 0.08 mL/min) and pathological ocular test results. None of the carrier SSA+ included in the preclinical SS group fulfilled the classification criteria for pSS. A control group (*n* = 8) comprised age-matched and gender-matched healthy subjects with no subjective symptoms of ocular or oral dryness. At the time of saliva sampling, no patients or controls had a diagnosis of oral periodontitis. The study was approved by the Medical Research Ethics Committee of the University of Pisa. All the patients provided an informed consensus to participate in the study. 

### 2.2. Salivary EV Enrichment and Sample Preparation for Mass Spectrometry-Based Proteomics

After refraining from eating and drinking for at least 8 h, volunteers were asked to rinse their mouths with water to remove food debris. Then, unstimulated saliva samples were collected by spitting directly into a 50 mL sterile centrifuge tube for 5 min. The saliva samples of pSS patients, SSA+ patients, and healthy controls were enriched in extracellular vesicles (EVs) by differential centrifugation. In detail, each sample was centrifuged at 300× *g* to eliminate cells, bacteria, and potential food debris. The supernatant was centrifuged at 2000× *g* for 30 min at 4 °C to remove apoptotic blebs and was then subjected to ultracentrifugation at 100,000× *g* for 2 h at 4 °C to obtain EV-enriched saliva samples from each subject [16]. Isolated vesicles (Appendix A) are a mixture of three differently sized distributions (69 ± 24 nm, 330 ± 150 nm, and 5307 ± 395 nm, with relative intensities of 61.7%, 33.7%, and 2.9%, respectively). 

Proteins were extracted with 50 mM ammonium bicarbonate (Merck, Darmstadt, Germany) and 1% sodium deoxycholate (Sigma-Aldrich, St. Louis, MO, USA). Briefly, samples were lysed through five repeated freeze-thaw cycles (freezing in liquid nitrogen for 30 s and thawing at 50 °C for 2 min), sonicated with MSE Soniprep 150 (MSE Scientific Instruments, Nuaillé, France) for 5 min (five cycles of 20 s, with an interval between cycles of 40 s on ice) and then clarified by centrifugation at 16,000× *g* for 10 min at 4 °C. Protein concentration was determined by the bicinchoninic acid assay (Thermo Scientific, Rockford, IL, USA) using serum albumin as standard. For each condition, 100 µg of proteins were reduced with dithiothreitol (10 mM for 30 min, at 65 °C) and alkylated using iodoacetamide (22 mM for 30 min, at room temperature) in dark conditions (both reagents from GE Healthcare, Chicago, IL, USA). Protein digestion was performed using trypsin (*w*/*w* ratio 1:50, GE Healthcare, Chicago, IL, USA) at 37 °C for 16 h. The samples were incubated with 1% trifluoroacetic acid (Sigma-Aldrich, St. Louis, MO, USA) for 45 min at 37 °C to quench the trypsin reaction and to remove sodium deoxycholate by acid precipitation. The samples were centrifuged at 16,000× *g* for 10 min and, subsequently, desalted with Mobicol spin columns (Mo Bi Tec, Goettingen, Germany), equipped with 10 µm pore-size filters and filled with VersaFlash C18 spherical 70 Å silica particles (Supelco Analytical, Bellefonte, PA, USA). The peptides were lyophilized and were, consequently, dissolved in 2% acetonitrile/0.1 formic acid to achieve a final peptide concentration of 2 µg/µL before high-performance liquid chromatography–tandem mass spectrometry (HPLC-MS/MS) analysis.

### 2.3. MS Acquisitions: DDA- and DIA-Based SWATH-MS

The equivalent of 10 µg per sample was directly loaded with an Eksigent expert™ microLC 200 system and acquired using a 5600+ TripleTOF mass spectrometer (both ABSciex Concord, ON, Canada). After loading, the peptides were separated on a Jupiter 150 × 0.3 mm, 4 µm 90 Å capillary column with a gradient from 5 to 22% of buffer B (acetonitrile/0.1 formic acid) in 50 min at a flow rate of 5 µL/min. The LC column was directly interfaced with a DuoSpray™ ESI ion source, operating at 5.5 kV, for peptide ionization. We used data-dependent acquisition (DDA) tandem mass spectrometry for spectral library generation, which relies on a survey MS1 scan, followed by the selection of a maximum of 20 of the most abundant precursor ions, and their further fragmentation by collisional induced dissociation (CID), using nitrogen N_2_ as an inert gas to generate MS2 spectra. The MS1 survey and MS2 scans were acquired with a resolving power of 30,000 and 25,000 and over a mass range of 250–1250 *m*/*z* and 150–1500 *m*/*z*, respectively. The isolation width for precursor ion selection was set at 0.7 *m*/*z* on a Q1. The accumulation time was set to 250 milliseconds for MS1 scans, while it was set to 100 milliseconds for MS2 scans. Charge states of 1+ were excluded from ion selection. Rolling collision energy with a collision energy spread across 5 eV and background subtraction were enabled to achieve the optimal fragmentation, according to the *m*/*z* ratio and charge state, and to increase sensitivity. 

EV-enriched saliva peptide samples from pSS patients, SSA+ patients, and healthy controls (*n* = 27) were analyzed using a data-independent method based on the sequential window acquisition of all the theoretical fragment ion spectra (SWATH-MS). SWATH acquisitions were performed over 40 overlapping isolation mass windows of variable width (min width 10 Da, mass selection with 1 Da of window overlap), depending on the peptide density distribution along the entire mass range of 250–1250 *m*/*z* (Appendix A). Precursor ion activation was performed by CID, as described before. An accumulation time of 50 milliseconds for MS1 and 80 milliseconds for MS2 scans resulted in an overall duty cycle of 3 s (≈8 points per elution peak).

### 2.4. Spectral Library Generation and Protein Quantification

The spectral library was created by combining the outputs from DDA-MS runs of three pools obtained by putting together different samples of the three groups: pSS patients, SSA+ patients, and healthy controls. Two replicate injections were performed for each pool and the acquired raw data (.wiff, profile spectra) were processed in parallel with two pipelines, in order to increase the coverage of the library.

In detail, DDA-MS data were first processed through the SCIEX proprietary pipeline, using ProteinPilot™ software (ABSciex, Concord, ON, Canada). Iodoacetamide was selected as the Cys alkylation agent, with trypsin as the digestion enzyme. Spectra were searched against a reviewed human database (UniProtKB/Swiss-Prot, 20,381 sequences, release February 2021), using as the detected protein threshold confidence greater than 10%. Only proteins identified with global FDR from a fit lower than 1% were included in the library.

Simultaneously, raw data were converted into a peak list format (.mzML, centroid spectra) using the ProteoWizard tool, qtofpeakpicker [17]. Protein identification was carried out using both the X!Tandem and Comet search tools through the Trans-Proteomic Pipeline (TPP) software suite [18], to increase the robustness of identification. Peak lists were searched against the previously reported human database, using a precursor ion and fragment ion tolerance of 20 and 50 ppm, respectively, a precursor charge state between 2+ and 5+, and a maximum number of 2 trypsin missed cleavages. Carbamidomethylation (+57.021 Da) of the cysteine residues and the oxidation (+15.995 Da) of methionine residues were chosen as fixed and variable modifications, respectively. Peptide spectrum matches (PSMs) from both the X!Tandem and Comet search tools were scored, combined, and re-scored using PeptideProphet and iProphet to increase the confidence between correct and incorrect hits. False positives were filtered out using a protein FDR lower than 5%, using MAYU software (corresponding to the lowest iProphet probability of 0.90); the resulting list of PSMs was converted into a redundant spectral library, using SpectraST [19]. At this stage, the retention times of all the peptides were transformed into normalized retention times using 11 endogenous retention-time reference peptides (iRTs), which cover the entire retention time range of the analyzed samples. MS2 spectrum entries corresponding to a redundant peptide sequence assignment were collapsed into a single consensus spectrum, to increase the accuracy and consistency of the fragmentation pattern. A consensus of the spectral library was then converted into a SWATH assay library, in which the following parameters were included. A minimum number of 6 fragment ions per peptide precursor and fragment ions smaller than 350 and bigger than 2000 *m*/*z* were filtered out; only b- and y-ion types were considered, and only fragments with a mass accuracy equal to or below ±0.05 Th of the expected mass were used. 

The two assay libraries (in table format) were separately uploaded into the PeakView program (AB Sciex, version 2.2) and the retention time alignment to SWATH-MS samples was obtained using selected peptides (top confidence and top-level transitions) from the top-scoring protein (Appendix A). Libraries with updated retention times were joined together in a single library, then this library was used for SWATH-MS quantification, performed using PeakView with SWATH™ Acquisition MicroApp 2.0. The processing settings were: 15 peptides per protein, 7 transitions per peptide, 95% peptide confidence, and 5% FDR. The XIC options were an extraction window of 8 min and a width of 50 ppm. Fragment ion signals/peptide precursor groups (transitions) areas were extracted and integrated together to obtain the peptide abundances. Protein abundances were calculated by summing up the abundances of their specific peptides.

### 2.5. Data Processing and Statistical Analysis

Experimental data were analyzed using R software (version 4.0.3). The protein abundances of pSS, SSA+, and healthy subjects were normalized, based on the media total abundance. First, to reveal patterns in the data, an unsupervised multivariate method was used. Principal component analysis (PCA) was performed to reduce the variation of the protein dataset to a small number of dimensions. 

Then, a univariate analysis was conducted. For each protein fold change (FC), the value was calculated for the pSS and SSA+ groups as the ratio between the mean of expression in patients and the mean of expression in the control group. Proteins were considered to be differentially expressed when the fold changes were higher than 1.5 (FC ≤ 1/1.5 or FC ≥ 1.5). Proteins were considered of interest when they showed a concordant differential expression for both the pSS and SSA+ groups. A Shapiro–Wilk test was used to assess how close the data were to a normal distribution. As the data were not normally distributed (*p*-values < 0.05), a two-tailed Mann–Whitney U-test was performed in order to consider the significant differences in protein levels between patients and controls. Proteins were considered significant when exhibiting a *p*-value lower than 0.05.

### 2.6. Functional Network Analysis

The search tool for the retrieval of interacting genes/proteins (STRING) [20] was used to explore how concordant, differentially expressed proteins in the pSS and SSA+ groups were interrelated to form protein–protein interaction networks. This was achieved by applying as the active interaction sources the previous experiments, databases, text mining and co-expression, and medium confidence. These proteins of interest were also subjected to gene ontology (GO) analysis. GO analysis was conducted using the BiNGO plug-in [21] of the Cytoscape software (version 3.8.2), which allows the assessment of the enrichment of a GO term in a test set of proteins. Enrichment analysis was performed using a hypergeometric test and the resulting *p*-values were corrected with the Benjamini–Hochberg procedure. GO terms with a *p*-value < 0.01 were considered significant for the analysis. The ontology and annotation files used were the go.obo file (version 1.2, released on 1 February 2021) and the goa_human.gaf file (version 2.1), respectively, available at the GO website (http://geneontology.org/ accessed on 1 April 2021). Three separate analyses were conducted with respect to biological processes (BPs), molecular functions (MFs), and cellular components (CCs). To better understand the results of the BPs analysis, the numerous GO terms in the output were clustered using the AutoAnnotate plug-in of the Cytoscape software (using the GLay clustering algorithm), to group together those biological processes that shared common proteins.

## 3. Results

### 3.1. EV-Enriched Saliva Protein Profiling of pSS, SSA+, and Healthy Control Subjects

Eleven pSS patients, 8 SSA+ carriers, and 8 age- and sex-matched healthy volunteers were included in the study. The patients’ main clinical features are summarized in Table 1. The SSA+ carriers with pre-clinical SS were significantly younger compared to those presenting SS. No further differences were detected between the groups. Comparison between pSS patients, carrier SSA+ subjects, and healthy controls was carried out using the normalized protein abundances derived from the integration of all peptide-specific extract ion chromatograms for each matched protein. The proteins were quantified across samples using the ion spectral library previously built through two complementary approaches: the SCIEX proprietary pipeline and the Trans-Proteomic Pipeline, which allowed us to quantify 145 proteins among the 161 identified and 131 proteins among the 178 identified, respectively (106 quantified proteins were in common between the two approaches). In this way, a total of 166 unique proteins were quantified across all EV-enriched saliva samples.

PCA was used as an unsupervised exploratory technique to reduce protein data complexity and visualize how those samples belonging to different groups correspond to each other according to their EV-enriched saliva protein profile. First two principal components, PC1 and PC2 (as shown in Figure 1), explain the 19.73% and the 12.53% of the data variance, respectively. Although the first component is the one that contains a greater percentage of variance among the data, it is the second that is of particular interest in discriminating the two groups of patients from the control group. The different groups are not completely separated; this could be due to those components of variability explained by interindividual differences, combined with the relatively low number of samples. Nevertheless, according to PC2, it is possible to observe that the pSS patients differed from controls and, more interestingly, that SSA+ patients were closer to the pSS group than they were to the control group.

### 3.2. Concordant Differentially Expressed Proteins in pSS and SSA+ Patients

Data analysis of the expression level of 166 quantified proteins was conducted with the aim of highlighting those proteins with an expression trend in pSS patients and carrier SSA+ that was different from the control group and, at the same time, similar to each other. This led to the selection of 57 concordant differentially expressed proteins: 28 with a FC higher than 1.5 for both the pSS and the SSA+ groups, compared to the control group, and 29 with a FC lower than 1/1.5 for the two groups. The concordant expression pattern of these 57 proteins (Appendix A) represents the common protein signature of the EV-enriched saliva of the pSS and SSA+ samples. 

We focused on these proteins to investigate in more detail what physio-pathological mechanisms could link the two groups of patients.

### 3.3. Functional Network Analysis of pSS and SSA+ Patients’ Common Protein Signatures

STRING was used to explore how these concordant differentially expressed proteins in pSS and SSA+ patients are interrelated to form protein-protein interaction networks. The STRING database was able to identify only 44 out of the 57 proteins; it is worth noting that all of the 13 non-recognized proteins belonged to immunoglobulines. Beyond this, the protein–protein interaction network built on the 44 identified proteins results in a number of edges of 91 and an average node degree of 4.14 (PPI enrichment *p*-value < 1.0 × 10^−16^), showing that these proteins are strongly interrelated to each other. Using MCL as the clustering method, it was possible to highlight the presence of various functional clusters, as shown in Figure 2, including: typical acinar salivary proteins (MUC5B, PIP, CST4); proteins involved in the protease/antiprotease balance (cystatins); constituents of the cytoskeleton (several keratins, including KRT6A, which is involved in wound repair); proteins involved in cell metabolism, cell differentiation, proliferation and apoptosis; histone proteins (HIST1H2BD, HIST1H1B, HIST1H1E, HIST1H4F, H2AFJ), the structural components of chromatin, which plays a key role in the regulation of transcription; and finally, among the best-represented proteins, we found inflammatory and immunity-related proteins.

GO enrichment analysis was carried out with respect to biological processes (BPs), molecular functions (MFs), and cellular components (CCs), to evaluate the functional networks in concordant differentially expressed proteins in pSS and SSA+ patients. Unlike before, the GO database made it possible to annotate the entire set of 57 proteins. The results showed significant enrichment of BP GO terms associated with the activation of the immune system, particularly of humoral immunity, with the activation of B lymphocytes and secretion pathways, as well as innate immunity, with the activation of inflammatory processes (Appendix A). The most significantly enriched MFs were antigen- and immunoglobulin-receptor binding, together with the endopeptidase regulator activity, while for CCs, as expected, extracellular regions emerged, with about half of the differential expressed proteins being attributable to extracellular vesicles. A high significance also resulted for the immunoglobulin complexes (Appendix A).

### 3.4. Assessment of Specific Proteins of Interest in Sjögren’s Syndrome

After completing a functional characterization of the common protein signature of the EV-enriched saliva of pSS and SSA+ samples, a subset of these proteins was investigated in deeper detail. Among the 57 proteins, 19 are significantly dysregulated in pSS patients compared to controls, while 18 are significantly dysregulated in SSA+ patients compared to controls, with a FC higher than 1.5 or lower than 1/1.5 and the statistical significance set at *p* < 0.05. Interestingly, 11 of these significantly dysregulated proteins are in common, as shown in Figure 3a: the fibrinogen gamma chain, different immunoglobulins chains, keratin KRT6A, prolactin-inducible protein, cystatin-SA, lipocalin-1, and the HLA class-I histocompatibility antigen (alpha chain G).

Finally, four proteins were selected for their biological interest in pSS. The expression levels of these proteins were evaluated in the three different groups of subjects, healthy controls, SSA+, and pSS patients (Figure 3b), to fully characterize their dysregulation between patients and controls. Here, the up-regulation of lactotransferrin and down-regulation of cystatin-SA, lipocalin-1, and prolactin-inducible protein emerged as the most significant dysregulation, able to bring closer the EV-enriched saliva protein profiling of SSA+ patients with those of pSS patients.

## 4. Discussion

In this pilot study, we analyzed the salivary proteomic profiles of patients with established SS and with pre-clinical SS, identifying a common protein signature in their EV-enriched saliva. We found that several inflammatory, immunity-related, and typical acinar proteins are expressed differently in pSS and in pre-clinical SSA+ carriers compared to healthy controls, suggesting that saliva may closely reflect exocrine gland inflammation from the early phases of the disease. Indeed, this study confirms the value of EV-enriched salivary proteomics for the identification of reliable biomarkers for SS, considering that several of the identified biomarkers have already been recognized in the whole saliva of patients with established SS. To date, several proteomic studies have been conducted using the whole saliva of pSS patients [22], mostly in comparison with healthy volunteers or subjects with non-immune-mediated sicca symptoms. Recently, the proteomics of salivary extracellular vesicles has been characterized by Aqrawi et al. [9]. The upregulation of proteins involved in innate immunity (LCN2), cell signaling (CALM), and wound repair (GRN and CALML5) were detected in the saliva of pSS patients. The salivary EVs also displayed biomarkers critical for the activation of the innate immune system (SIRPA and LSP1) and adipocyte differentiation (APMAP). Overall, as reported by Cecchettini et al. [10], most of the data have highlighted that the pSS salivary proteome is characterized by a decreased expression of acinar proteins that are physiologically involved in oral mucosa healing and protection, lubrification, digestion, the sense of taste, and dental mineralization. In contrast, a number of inflammatory proteins have been described as over-expressed, as well as immune-related molecules (i.e., immunoglobulins and beta-2 microglobulin).

However, to the best of our knowledge, this is the first study exploring the potential value of saliva as a source of early SS biomarkers that could be identified even in the preclinical phase of the disease, as represented by carrier SSA+ patients. Of note, among these putative early biomarkers, there are typical acinar proteins, such as MUC5B, PIP, CST4, and lipocalin 1, suggesting that proteins specific to the salivary and lacrimal glands may serve as novel biomarkers for SS. These results are partially in line with several recent studies that have suggested the use of novel autoantibodies to proteins that are specific to the exocrine glands (SP-1 (salivary gland protein-1), PSP (parotid secretory protein), and CA-6 (carbonic anhydrase VI)) as novel biomarkers for SS that may appear earlier in the course of the disease [23]. Proteins like PIP, MUC5B, S-type cystatins, and lipocalin-1 play important roles in the salivary secretion processes and in saliva rheology, and their reduction may reflect an initial dysfunction of the salivary glands [24]. PIP has been linked to aquaporin channel formation. Gallo A. et al. confirmed, using different complementary omics techniques, the potential role of PIP as a novel biomarker for pSS, showing that PIP expression correlates with both the salivary flow rate and the minor salivary gland biopsies’ focus score. Finally, immunohistochemistry confirmed that PIP staining was faint in the mucus acini, while real-time PCR showed that PIP mRNA was significantly lower in pSS patients when compared to both no-SS sicca subjects and healthy controls, thus supporting the hypothesis that the observed reduction of PIP in pSS saliva may be related to a decrease in that protein’s production [25]. Mucin 5 B is a major contributor to the lubricating and viscoelastic properties of whole saliva, while lipocalin plays a role in taste reception, and S-type cystatins have antibacterial and antiviral activity, which is consistent with a protective function [26,27,28]. Analogously, we found that lactoferrin, a globular protein with antibacterial activity, was significantly upregulated in both established and pre-clinical SS [29]. Overall, these proteins may not only play a role in the early diagnosis of pSS but also represent ideal targets for novel therapeutic interventions aimed at restoring salivary function and rheology when curative interventions can, ideally, still be possible.

Our study has some limitations. The cohort is relatively small and only two patients of the carrier SSA+ group underwent a minor salivary biopsy. However, patients were accurately selected and matched to minimize the impact of patients’ demographic features on the results. Moreover, we precisely assessed the salivary and lachrymal functions of SSA+ carriers, which were well-preserved in all the cases. Another important limitation is the nature of the investigated samples since ultracentrifugation allows the recovery of EV-enriched saliva samples in which mucin, amylase, and other salivary proteins are predominant and can still mask other low-expressing proteins. The interference of highly abundant viscous proteins that are present in saliva makes EV isolation challenging [30]. However, ultracentrifugation is currently the gold standard method for the isolation of EVs from biological fluids [7], but the complexity of salivary fluid would need more proper and suitable EV isolation methods. 

Globally, our findings are consistent with the results of previous studies showing that in SS, an autoimmune-mediated insult of the salivary glands may occur several years before the onset of symptoms and much longer before the diagnosis. Indeed, our preliminary findings could pave the way for early therapeutic interventions and drug development processes in pSS.

## Figures and Tables

**Figure 1 biomolecules-12-00738-f001:**
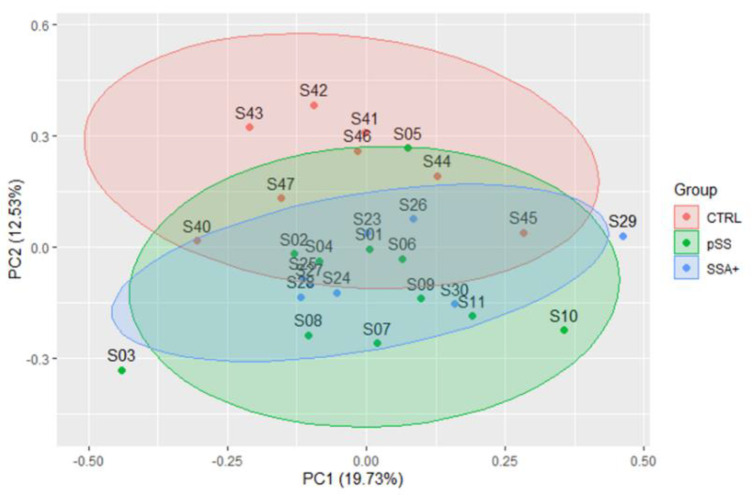
PCA scores plot of the first two principal components. Different groups of subjects are shown by colors: healthy controls in red, pSS patients in green, and SSA+ patients in blue. The ellipse around each group represents the 95% confidence interval, using Hotelling’s T2 statistics.

**Figure 2 biomolecules-12-00738-f002:**
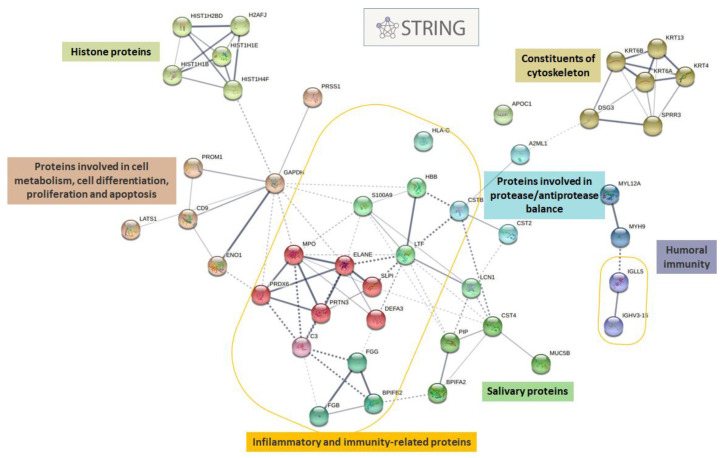
STRING protein–protein interaction networks of concordant differentially expressed proteins in pSS and SSA+ patients. The network shows the potential interactions of proteins inferred from experiments, databases, text mining, and co-expression with medium confidence. Different clusters, generated through the MCL clustering method, are indicated by color and the thickness of the connecting lines indicates the confidence of interactions.

**Figure 3 biomolecules-12-00738-f003:**
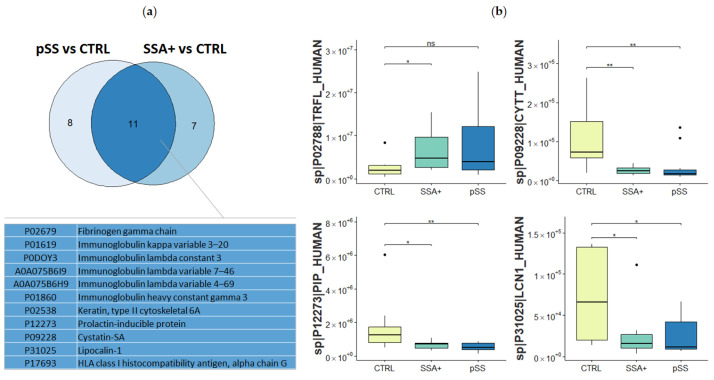
(**a**) Venn diagram of significantly dysregulated proteins (with a FC higher than 1.5 or lower than 1/1.5 and with statistical significance set at *p* < 0.05) in pSS and SSA+ patients, compared to controls. In total, 19 proteins were significantly dysregulated in pSS patients, while 18 proteins were significantly dysregulated in SSA+ patients, and 11 of these proteins are in common. (**b**) Box plot reporting protein abundances for proteins of biological interest in Sjögren’s syndrome. Protein abundance distributions are calculated for the three groups of subjects—healthy controls, SSA+, and pSS patients—and a two-tailed Mann Whitney U-test was performed to consider the significant differences in protein levels between patients and controls (statistically significant *p*-values are shown as follows: * *p*-value < 0.05, ** *p*-value < 0.01).

**Table 1 biomolecules-12-00738-t001:** Characteristics of the patients who presented a positive result for anti-Ro/SSA antibodies but had no sicca symptoms (SSA+ carrier) and primary Sjögren’s syndrome (pSS) patients. Continuous variables are presented as mean ± standard deviation (SD), while categorical variables as numbers and relative percentages.

	SSA+ Carrier (8)	pSS (11)	*p*-Value
Age (years, SD)	35 (8)	60 (8)	*** <0.001
USFR ^†^ (mL/min)	1.6 (1.9)	2.2 (1.8)	0.466
Schirmer’s (mm)	12 (5.5)	7 (6)	0.063
ESSPRI ^‡^	4.7 (2.7)	4.9 (3.6)	0.890
SSA	8/8 (100%)	7/11 (64%)	0.060
SSB	3/8 (37.5%)	1/11 (9%)	0.255

^†^ USFR, unstimulated salivary flow rate. ^‡^ ESSPRI, EULAR: Sjögren’s syndrome patient-reported index. *** *p*-value < 0.001

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
