# Peer review of "Salivary Proteomics Markers for Preclinical Sjögren’s Syndrome: A Pilot Study"

_biomolecules, 2022, doi:10.3390/biom12060738_

Round 1

Reviewer 1 Report

The authors have attempted to address the concerns raised by the reviewer.

The SWATH-MS approach is appreciated. However, the authors do not provide the Qc data for their SWATH analysis.

  1. Please provide the correlation matrix for intensities across biological and technical replicates of all the samples analyzed.
  2. Please provide the reproducibility data (%CV) for all transitions used for quantifying subsequent peptide and protein levels.
  3. What is the average transition intensity and how they are varied across runs?
  4. Could you please analyze your SWATH data to assess the sensitivity (ppm level abundance) of the optimized method used in this manuscript?
  5. How many transitions were used for a single peptide quant? How did the authors deal with overlapping transitions?
  6. Please depict the TIC reproducibility of all the DIA-MS runs.
  7. What is the RT variation between library (DDA) run and DIA runs?
  8. How many technical replicates were performed for each biological sample?

Author Response

We thank the reviewer for the opportunity to revise our manuscript. Our point-by-point response to the reviewer’s comments is in the attached PDF.

Reviewer 2 Report

Following minor grammatical corrections, I believe this manuscript is suitable for publication in Biomolecules. 

Author Response

We thank the reviewer for his/her comment. The paper has been carefully revised to fix grammar errors and typos. We hope that in this new form the manuscript can be accepted for publication in Biomolecules.

Round 2

Reviewer 1 Report

Accepted

This manuscript is a resubmission of an earlier submission. The following is a list of the peer review reports and author responses from that submission.

Round 1

Reviewer 1 Report

Giorgi et al. identified the potential salivary extracellular vesicles biomarkers in pre-disease stage of primary Sjögren’s syndrome by using mass spectromety-based proteomics. The manuscript is well written. A few comments are listed below.

  1. Healthy controls were included. However, are the identified protein biomarkers specific to primary Sjögren’s syndrome? Should disease controls be included (e.g. SLE)?
  2. When the saliva samples were collected, were the patients treated with immunosuppression therapy?
  3. There are some literatures on the salivary biomarkers in this disease. Have the authors compared with those previous studies?
  4. In the methods, what sonicator was used (line 78)?

Author Response

Giorgi et al. identified the potential salivary extracellular vesicles biomarkers in pre-disease stage of primary Sjögren’s syndrome by using mass spectromety-based proteomics. The manuscript is well written. A few comments are listed below.

1. Healthy controls were included. However, are the identified protein biomarkers specific to primary Sjögren’s syndrome? Should disease controls be included (e.g. SLE)?

We thank the reviewer for this important consideration that gives us the opportunity to underline some key points. As the reviewer correctly highlighted, the identified protein biomarkers are not specific for pSS. However they have been extensively associated in the literature to pSS-related glandular inflammation and dysfunction. In this paper our main aim was to differentiate pSS from SSA+ carrier. Preclinical SS presents anti-Ro/SSA positivity but no sicca symptoms and did not fulfil the classification criteria for SS (carrier SSA+). In this study we particularly focused on “preclinical Sjögren” (defined as asymptomatic carriers of anti-SSA antibodies) and we included Sjögren’s patients as unhealthy control samples and healthy control samples as “background” to demonstrate a consistent trend of marker expression from controls to overt Sjogren. We agree that future studies including patients with SLE or other connective tissue diseases sharing similar auto-antibodies profiles will help to identify common pathways underlying complex phenotypes of autoimmunity.

2. When the saliva samples were collected, were the patients treated with immunosuppression therapy?

None of the SSA+ carrier was under immunosuppression therapy. Patients with pSS were on stable doses of hydroxychloroquine (200 mg/die).

3. There are some literatures on the salivary biomarkers in this disease. Have the authors compared with those previous studies?

According to the reviewer suggestion, references regarding salivary biomarkers in primary Sjögren’s Syndrome were included in Discussion section. There are papers from many authors and also our previous papers on Sjogren syndrome and salivary  markers (i.e. Cecchettini, Antonella, et al. "Phenotyping multiple subsets in Sjögren’s syndrome: a salivary proteomic SWATH-MS approach towards precision medicine." Clinical proteomics 16.1 (2019): 1-11), but there are not papers on salivary proteomics in no-SS sicca cases (SSA+) that are the novelty of this paper.

4. In the methods, what sonicator was used (line 78)?

EV-enriched saliva samples were sonicated for 5 min (five cycles of 20 sec with an interval between cycles of 40 sec on ice) using MSE Soniprep 150.

Reviewer 2 Report

  1. The introduction is insufficient to introduce the work. Some of the introduction is a reiteration of the abstract.  Specifically, EVs have not been introduced anywhere in this manuscript.  Readers need to know what EVs are, briefly about the body of work on EVs in SS, and why you enriched the saliva for EVs. Also, the proteomic analyses you used – and that have been used by others in this field – should be introduced. 
  2. There are several grammar and spelling errors that need to be addressed prior to publication
  3. This line in the discussion needs to include references and be discussed in much greater detail: “Indeed, this study confirms the value of EV salivary proteomics for the identification of reliable biomarkers for SS, considering that several of the identified biomarkers have been already recognized in whole saliva of patients with established SS.” You have not discussed the value of EV salivary proteomics in this manuscript nor incorporated your findings into the body of work.
  4. “However, to the best of our knowledge this is the first study exploring the potential value of saliva as a source of SS early biomarkers that could be identified even in a preclinical phase of the disease.” There are other publications that have looked at saliva and salivary EVs as biomarkers for early SS diagnoses. Some of the studies in these papers should be discussed and referenced. I’ve included a couple papers below that cite several relevant publications. 
    • https://arthritis-research.biomedcentral.com/articles/10.1186/s13075-017-1228-x
    • https://www.ncbi.nlm.nih.gov/pmc/articles/PMC4955829/
  5. A more detailed discussion including the relevance of some of the identified biomarkers should be included.

Author Response

  • The introduction is insufficient to introduce the work. Some of the introduction is a reiteration of the abstract.  Specifically, EVs have not been introduced anywhere in this manuscript.  Readers need to know what EVs are, briefly about the body of work on EVs in SS, and why you enriched the saliva for EVs. Also, the proteomic analyses you used – and that have been used by others in this field – should be introduced. 

According to the reviewer suggestion, introduction section was revised. More details and references regarding EVs, EVs in primary Sjögren’s Syndrome and proteomics technique used in our study are provided.

  • There are several grammar and spelling errors that need to be addressed prior to publication

We thank the reviewer for that observation, we revised the entire manuscript to fix grammar and spelling errors.

  • This line in the discussion needs to include references and be discussed in much greater detail: “Indeed, this study confirms the value of EV salivary proteomics for the identification of reliable biomarkers for SS, considering that several of the identified biomarkers have been already recognized in whole saliva of patients with established SS.” You have not discussed the value of EV salivary proteomics in this manuscript nor incorporated your findings into the body of work.

We thank the reviewer for his/her remark. Here we would point out that pSS dysregulated proteins determined in this work had also emerged from previous studies. A more detailed discussion was provided. In addition, we replaced “EV salivary proteomics” with “EV-enriched salivary proteomics” that is more in line with our samples. This kind of samples was selected with the aim of simplify the salivary proteome dynamic range, reducing highly abundant proteins that can mask other pivotal but less represented proteins. One of the greatest challenges in analyzing the salivary proteome is the wide range of concentration of different proteins. In this scenario the possibility of exploring extracellular vesicles (EVs)-associated proteins has represented a possible answer in the search for disease specific biomarkers. The value of EV salivary proteomics was discussed in the revised version of the manuscript (Introduction), but it is beyond the scope of our work, which is rather to analyze and compare salivary proteomics of patients with established pSS and patients with pre-clinical SS (carrier SSA+) identifying a common protein signature in their salivary fluid.

  • “However, to the best of our knowledge this is the first study exploring the potential value of saliva as a source of SS early biomarkers that could be identified even in a preclinical phase of the disease.” There are other publications that have looked at saliva and salivary EVs as biomarkers for early SS diagnoses. Some of the studies in these papers should be discussed and referenced. I’ve included a couple papers below that cite several relevant publications. 
  • https://arthritis-research.biomedcentral.com/articles/10.1186/s13075-017-1228-x
  • https://www.ncbi.nlm.nih.gov/pmc/articles/PMC4955829/

We agree with the observations of the reviewer, references regarding salivary biomarkers in primary Sjögren’s Syndrome were included in Discussion section. However, in the sentence highlighted by the reviewer “preclinical phase of the disease” referred to the proteomics analysis of  patients with preclinical SS that presented anti-Ro/SSA positivity but no sicca symptoms and did not fulfill the classification criteria for SS (carrier SSA+), that is the main focus of the work.

  • A more detailed discussion including the relevance of some of the identified biomarkers should be included.

We thank the reviewer for his/her suggestion. A paragraph has been added in Discussion section.

Reviewer 3 Report

Please address the following:

  1. How much saliva was collected?
  2. What was the SoP of saliva collection and storage?
  3. The discovery phase analysis was performed <20 samples. What is the statistical power  here?
  4. Please discuss the normal salivary proteome with respect to your identified protein nos in saliva?
  5. Please validate a few candidate markers to show robustness in an independent cohort.

Author Response

Please address the following:

1. How much saliva was collected?

For each subject involved in the study unstimulated saliva sample was collected spitting directly into a 50 ml sterile centrifuge tube for 5 min. Between 1 and 7.5 mL of saliva was obtained from each pSS subject whereas volumes obtained from controls ranged from 2.5 to 10 mL.

2. What was the SoP of saliva collection and storage?

After refraining from eating and drinking for at least 8 h, volunteers were asked to rinse mouth with water to remove food debris. Then, unstimulated saliva samples were collected spitting directly into a 50 ml sterile centrifuge tube for 5 min. To minimize degradation of the proteins, fresh saliva samples were immediately processed and kept on ice during the process. Freshly collected saliva were centrifuged at 300g for 30 min at 4 °C to remove cells, bacteria, and potential food debris. The saliva supernatant was diluted in PBS and subsequently centrifuged at 2,000g for 30 min at 4 °C to remove apoptotic blebs. The supernatant was transferred in a polycarbonate centrifugation tube (26.3 mL capacity, Beckman Coulter) and used for ultracentrifugation using a Beckman Coulter XL-90 Ultracentrifuge. The tube was filled to the brim with PBS to prevent collapse during ultracentrifugation and labelled to mark the expected location of the pellet. Ultracentrifugation was performed at 100.000 x g for 2 h at 4 °C and, finally, the pellet was resuspended in 200 µl of 50 mM ammonium bicarbonate, 1% sodium deoxycholate and stored at -80°C until sample preparation for mass spectrometry analysis.

3. The discovery phase analysis was performed <20 samples. What is the statistical power here?

For the restricted number of patients, the statistical power exceeds 0.8 only for large effect-size (d > 1.5). However, the limited number of patients was mainly due to the nature of primary Sjögren’s syndrome, whose prevalence is 1-5 /10 000 (ORPHA:289390). This aspect was also addressed in the manuscript as a limitation of the study.

4. Please discuss the normal salivary proteome with respect to your identified protein nos in saliva?

The value of EV salivary proteomics was discussed in the revised version of the manuscript (Introduction), but it is beyond the scope of our work, which is rather to analyze and compare salivary proteomics of patients with established pSS and patients with pre-clinical SS (carrier SSA+) identifying a common protein signature in their salivary fluid.

Normal salivary proteome contains various highly abundant proteins such as amylase and immunoglobulin (IgG) that can mask other low-abundant proteins, such as extracellular vesicles (EVs)-associated proteins. EVs play a significant role in intercellular signalling and EV proteomics has been previously evaluated as a tool for the identification of novel biomarkers in pSS. EV-enriched saliva samples was selected for proteomics analysis of our cohort with the aim of simplify the salivary proteome dynamic range, reducing  highly abundant proteins that can mask other pivotal but less represented proteins. One of the greatest challenges in analyzing the salivary proteome is the wide range of concentration of different proteins. In this context it is likely that the concomitant analysis of whole saliva and EVs proteins may provide a comprehensive picture of the disease mechanisms offering the possibility of identifying novel specific biomarkers.

Specifically, comparing the EV-enriched saliva protein library used in the work (obtained putting together EV-enriched saliva samples of the three groups: pSS patients, SSA+ patients and healthy controls) with a protein library obtained from whole saliva (after removing cells, bacteria and potential food debris and diluting with PBS), we have an increase of 29.7% ((166-128)/128) of identified proteins. The vast majority (92/128 = 71.9%) of whole salivary proteins were identified also in EV-enriched saliva samples, but  44,6% (74/166) seems to be specific for this kind of sample, which thus leads to the possibility of exploring a more informative overview of the disease mechanisms.

(Please check the attachment for the figure)

5. Please validate a few candidate markers to show robustness in an independent cohort.

The study, as reported in the title, aims to represent a pilot study. The aim was to analyze and compare the salivary proteomics of patients with established pSS and patients with pre-clinical SS (carrier SSA+) to evaluate common aspects in their salivary fluid. Validation in an independent cohort represents our future perspective, but enrolling new patients is far from simple for this type of disease and therefore requires a dedicated study.

Round 2

Reviewer 1 Report

The authors have addressed all the comments.

Reviewer 3 Report

The proteomics findings with a small size of discovery phase must be validated in a larger cohort.